# Should I stay or should I go—Medical assistants´ experiences and coping with patient demand and lack of appreciation during the Covid-19 pandemic

**Anastasia Suslow**[1*], **Kathrin Schlößler**[1], **Nino Chikhradze**[1¤a], **Romy Lauer**[1¤b], **Michael Pentzek**[2], **Achim Mortsiefer**[3], **Horst Christian Vollmar**[1], **Ina Carola Otte**[1¤a]

**1** Institute of General Practice and Family Medicine (AM RUB), Medical Faculty, Ruhr University Bochum, Bochum, Germany, **2** Institute of Family Practice, Medical Faculty, University of Duisburg-Essen, Duisburg-Essen, Germany, **3** Institute of General Practice and Primary Care, Chair of General Practice II and Patient-Centredness in Primary Care, School of Medicine, Faculty of Health, Witten/Herdecke University, Witten, Germany

¤a Current address: Department of Health Services Research, Institute for Diversity Medicine, Ruhr University Bochum, Bochum, Germany
¤b Current address: Department of Medical Informatics, Biometry and Epidemiology (AMIB), Medical Faculty, Ruhr University Bochum
* anastasia.suslow@rub.de

## Abstract

### Background

This article explores the psychological burden experienced by medical assistants (MAs) in General Practices during the Covid-19 pandemic (Corona virus disease 2019 (SARS-CoV-2)) in Germany. The study aims on demanding patient behavior, increased workload, and the perceived lack of appreciation and discuss their potential impact on the MAs´ well-being and career decisions.

### Methods

A qualitative approach was utilized. MAs were included via a regional practice network as well as professional associations and newsletters. In total, 21 interviews with MAs from various federal states in Germany were conducted between April and September 2021. The semi-structured interview guideline focused on daily work challenges during the pandemic and its consequences. Interviews were recorded, transcribed, and analyzed using qualitative content analysis according to Kuckartz.

### Results

The findings highlight core challenges, including demanding communication with patients, lack of appreciation in the media, a high workload, resilience versus career migration, and the needs and wishes of MAs in their everyday work. Abusive language, insults, and theft of materials by patients added significant stress. The interviews reveal how important teamwork and a supportive working environment are for overcoming these challenges.

**Data availability statement:** All relevant data are within the paper and its Supporting Information files.

**Funding:** This study was financially supported by Ruhr University Bochum in the form of a research grant (F1003-2020), via FoRum (Research Funding of the Medical Faculty of the Ruhr University Bochum), received by the Institute of General Practice and Family Medicine (AM RUB), Ruhr University Bochum. No additional external funding was received for this study. The funder had no role in study design, data collection and analysis, decision to publish, or preparation of the manuscript.

**Competing interests:** I have read the journal's policy and the authors of this manuscript have the following competing interests: Kathrin Schlößler is an employed doctor in a general practitioner's practice from whose practice MA were interviewed for this study. As an employed GP, she also worked as a trial physician in a study under the German Drug Law (sponsored by Lilly). She received her regular salary as an employed doctor and received no compensation or expense allowance from the sponsor. This does not alter our adherence to PLOS ONE policies on sharing data and materials.

**Abbreviations:** Covid-19, Corona virus disease 2019 (SARS-CoV-2); MA, Medical Assistant; PIPER, Pandemic Management in General Practice - Experience and Reflection.

## Conclusions

The study underlines the urgent need for societal and political awareness regarding the challenges faced by MAs, especially during public health crises. The perceived social egoism in patient behavior, coupled with a lack of recognition and appreciation, contributed to a challenging work atmosphere and potential burnout risk. Recommendations include enhancing support for MAs, recognizing their contributions in the media, and fostering collaborative efforts between practitioners and policymakers to address the unique challenges in general practices.

## Trial registration

German Register of Clinical Studies (DRKS) DRKS00032402; https://drks.de/search/de/trial/DRKS00032402 (Registration Date: 14.08.2023)

## Background

During the Covid-19 pandemic (Corona virus disease 2019 (SARS-CoV-2)), the tasks and workload of all medical professions and facilities increased enormously.

While many studies explored the impact of the pandemic on hospitals, nurses and physicians in general, little focus was set regarding medical assistants (MAs) in General Practice and Family Medicine.

In Germany, MAs have a crucial role in General Practice. Usually, they are the first point of contact for patients [1]. Further tasks, depending on their role in practice, the scope of their vocational training and additional further qualifications, are practice management and patient care such as vaccination, taking blood samples and technical examinations such as electrocardiograms or lung function tests. However, unlike physician assistants who have graduated from university, they do not perform routine-consultations on their own [2].

During the peak of the Covid-19 pandemic from spring 2020 to summer 2021, the number of patients in general practices increased. The practices experienced high numbers of infections, and the workload of vaccinations against Covid-19 was added to the regular tasks of MAs. With MAs being figuratively also considered primary caregivers for the many patient concerns, MAs were facing many different stressors such as demanding patients, a high workload and high flexibility [3] and exposition to the risk of infection, specially before the vaccination was available. Indeed, MAs belong to the professions most frequently infected with Covid-19 [4].

The scope of the present PIPER study (Pandemic Management in General Practice – Experience and Reflection) was MAs' perception during the peak of Covid-19 in Germany. In another article, we highlighted the challenging organization of Covid-19 vaccinations on behalf of the MAs. In this article, we present partial results of this study on how MAs perceived patients' challenging behavior, how they dealt with it, and what consequences they drew.

## Methods

A qualitative approach in the form of interviews was chosen to enquire about all aspects of MAs´ experiences. The individual methodological steps and the researchers' approach are presented below.

## Recruitment

On April 1[st], 2021, we contacted 275 General practices associated with the network of the Institute of General Practice and Family Medicine (AM RUB) at the Ruhr University Bochum, North Rhine Westphalia, Germany. First, we contacted them in written form and then tried to contact them by phone. Because of the poor response rate (probably due to the high workload during the Covid-19 pandemic) we revised our recruitment strategy [5]. In June 2021, we advertised the study via newsletters from professional associations for MA training such as VERAH (Versorgungsassistentin in der Hausarztpraxis/ Supply assistant in general practice) and the General Practitioner Association, as well as via HAFO.NRW, which is a network of General Practitioner research practices [6,7]. In addition, we used snowball recommendations from already interviewed MAs. Inclusion criteria were professional employment in a general practice as a medical assistant (MA) or in practice management and willingness to participate. We planned to conduct approximately 25 interviews before assuming thematic saturation [8,9]. After 21 interviews on September 22[nd], 2021, it became apparent that the thematic saturation had already been reached, after the content was repetitive in the last interviews [8,9]. Therefore, the recruitment was concluded at this point.

## Data collection

We conducted the interviews using a semi-structured interview guideline allowing for natural course of the conversation. The guideline was based on a previous literature review [10] and focused on the daily work tasks of MAs. It was divided into four main topics: situation comparison, pandemic course, support, and outlook. Each topic contained keywords for further discussion (see Table 1). Additionally, the MAs were asked to fill out a short set of standardized questions from a questionnaire on socio-demographic characteristics (e.g., age, professional experience in years) and return it to the interviewer.

The interviews were conducted between April and September 2021 exclusively by telephone for reasons of participants' convenience, safety, and uncomplicated access by AS, a sociologist, with a focus on qualitative research and concise interviewing experience, in a personal conversation between interviewer and interviewee. On average, all interviews lasted 32 minutes with a range of 17 to 50 minutes. Throughout the interviews, the interviewer (AS) followed the semi-structured interview guideline and therefore was guided by the statements of the MAs and adapted the questions and the structure of the interview accordingly, which led to

**Table 1.  Main topics of the entire interview guideline.**

| Main topic | Keywords for further discussion |
|---|---|
| Situation comparison | • Current tasks<br>• Comparison to "before the pandemic"<br>• Procedure of the vaccination organization |
| Pandemic course | • Reflection on the start of the pandemic (in general)<br>• Everyday work at the beginning of the pandemic<br>• Changes in everyday working life during the pandemic |
| Support | • Availability of protective materials (face masks, overalls, gloves)<br>• Desired support<br>• Digital support (e.g., video consultation) |
| Outlook | • Further required actions<br>• Wishes for the future<br>• Possible measures for future pandemics |

a predominantly open and narrative interview. MAs did not know about the topics of the interview guideline beforehand. The MAs were granted the opportunity to talk about their feelings and experiences, which led to a high density of topics and statements. For this reason, we decided to consider the results in broad categories and to publish them according to focus. [11]. All interviews were audio recorded. An external transcription agency transcribed the interviews verbatim and pseudonymized them. The MAs did not receive any compensation for the interviews.

## Analysis process

We analyzed the interviews according to Kuckartz' qualitative content analysis [12] with the software MAXQDA [13]. The first three interviews were coded openly, these were then correlated with the results of the literature and discussed with other researchers to develop the subcategories and main categories. This research group met regularly to discuss open questions and finalize the coding process. Initially preliminary categories were derived deductively from the guideline. During the coding-process additional inductive categories emerged resulting in a final coding tree [12,14].

All interviews were coded using the final coding tree of AS (please see appendix Material); discrepancies were resolved in discussion with ICO, HCV, KS, and NC as supervisors according to a consensual approach. The authors are health scientists, nursing scientists or sociologists, General Practitioners, and medical ethicists with a qualitative focus and/or experience in analyzing interprofessional collaboration. The original quotes were translated from German into English by the authors. Translations were double checked by also translating them back from English into German.

## Ethical committee and consent

All participants received and signed an information sheet that included a privacy statement and an informed consent form. In addition, they had the opportunity to ask questions verbally or in writing. Written informed consent was obtained. A positive ethics vote of the Ethics Committee of the Medical Faculty of the Ruhr University Bochum has been obtained for this study (20-7010). All methods were performed in accordance with the relevant guidelines and regulations of this ethics committee. The study was conducted in accordance with the criteria of the Declaration of Helsinki.

## Results

A total of 21 MAs were interviewed by telephone over a period of six months. Table 2 summarizes sociodemographic and professional characteristics of the participants. Most participants were female covered a broad spectrum of age and professional experience.

**Table 2. Demographic variables of the interview participants.**

| Characteristics | |
| --- | --- |
| Gender n (%) | |
| Female | 20 (95) |
| Male | 1 (5) |
| Age mean (min-max) | 43 (26-61) |
| Professional experience in years, mean (min-max) | 22 (5-45) |
| Interview duration in minutes, mean (min-max) | 32 (17-50) |

## Main categories

During the analysis of the interviews, one main category (or focus) quickly evolved: "social egoism". We defined "social egoism" as a demanding and aggressive attitude among patients. In the following, we were able to identify four main categories in total. On the one hand, as just described above, one of the most significant categories of this manuscript: 1) demanding patients and social egoism, but also 2) (lack of) appreciation and high workload of MAs, 3) resilience vs. migration and 4) needs and wishes in everyday work, which are explained in more detail below. The following results represent the opinions of the MAs in direct quotes in quotation marks. The authors' interpretations are given before and after the quotes.

## Demanding patients and social egoism

During the pandemic, it seems that MAs made experiences with demanding patients and social egoism that frightened them and let them question their view of the world and humanity as a whole. Additionally, the MAs only had the current media (television, radio, newspaper, Internet) as a source available, as changes of the hygiene concept and regulations made by the government, or superior institutions were not passed on to them in a timely manner. This meant that they had the same level of information as their patients. Further stress was provoked by the governmental communication strategy. In Germany, stakeholder, e.g., politicians often disseminated their statements regarding regulations such as recommended vaccinations or priority in the media. The MA felt as "the last to know" as they often only had this media statements as a source for current practice recommendation:

> MA-11: "We sometimes already didn't know what to do in the morning. [...] [We had to] check by e-mail to see what new information was available [...]. It was constantly changing, the PCR tests, the rapid tests, how to invoice them, whom and what how to test. Who has to go into quarantine when entering from this country or that country? [...] A colleague of ours often arrived half an hour early and checked first: What's new today? How should we proceed? Because if the patients come and ask, we should at least know. And I really wish we'd always been given this information in advance so that we didn't have to deal with it ourselves."

However, statements (e.g., preference for vaccinating younger target groups) could not always be fulfilled, and the regulations were frequently amended and adapted. MAs find themselves in a difficult situation in which it is often difficult to meet patient expectations:

> MA-04: "So all these promises that came from the politicians, which then of course also went through the media, that the patients naturally thought 'Oh, huh? But that's what they said', and unfortunately, we now face the situation and have to say 'I'm sorry, we can't do it, unfortunately, it doesn't work the way they said it would'. We're basically right in the middle, this in-between part, and it's a bit difficult to deal with, so sometimes people say that they just don't understand."

MA-05 expressed concern that the pandemic has led to a change in values and with an increasing social egoism:

> MA-05: "There is no respect anymore, this disrespect. It's just this self-centeredness; people no longer look out for each other. [...] I know that humanity is changing, that's not a thing at all, but in such a short space of time, where everyone is just 'me, me, me' and they're just elbowing and-, I find that enormously frightening. And I don't think we're on the right

track. I could never have imagined that this human aspect, this thing that makes us human, this talking to each other or treating each other with respect, could fall by the wayside like this."

In some cases, MAs have also experienced personal insults. The behavior of some patients was considered inappropriate and difficult to deal with by the MAs:

MA-14: If someone says to me 'You stupid nut' or 'You old sod'; what I have to listen to or what we have to listen to, that is a personal form of address, of course, we take it personally. [...] And- and then you simply can't stop that. And then you can't be generous about it either. That's behavior that's not appropriate - and that's it. "

This was especially the case regarding vaccination. For MAs the perceived pressure regarding vaccinations was high:

MA-01: "So it's mostly about the vaccinations, people want to be vaccinated as soon as possible, no matter how old they are, or how ill they are. They want to be preferred, [...] sometimes you actually have to reprimand them, i.e., you have to end the conversation if the attitude doesn't change straight away. So [...] the tension is increasing in that regard. ".

One MA also reported that disinfectants and protective equipment had been stolen from the surgeries, leading to shortages. This forced the medical staff to take security measures:

MA-05: "And [...] we also had to make sure that we locked it away properly because the patients took disinfectant with them. They took masks with them, they took everything they could possibly need […]."

The challenging behavior of patients led to enormous psychological stress, which has built up, particularly over the first two coronavirus years 2020-2021.

Overall, the quotes show how communication with patients and increased expectations have become a significant stress factor for MAs.

## (Lack of) appreciation and high workload of MAs

Participants stated to not feel sufficiently appreciated. The MAs worked overtime during the pandemic, often far beyond their regular working hours:

MA-07: "None of us managed to say 'Well, work is over, I'm going home now, it's my right'. No, the waiting room was packed and the patients had to be looked after. (-) And we also sympathized with some of the patients who were fighting to survive. And when the wife sat there and cried because she thought her husband wouldn't come back from ICU, you don't say 'Well, I'm off to buy toilet paper.' Yes, of course, you listen to what the poor woman has to say. "

MA-08: "I've often said that I do something completely different, my job has become a completely different one, yes. [...] For example, I've also increased my job, I used to work 25 hours, now I work full time."

During the interviews it was emphasized that there was an imbalance in the distribution of appreciation and media and public attention:

> MA-07: "Everyone was talking about nurses. 'The poor ones' and 'They're saving the country' in other words 'single-handedly'. Nobody thought about the medical assistants in general practices. Nobody said a word about the fact that we sometimes worked 14 hours! Quite the opposite: they said, 'What, you're already getting off work? "

Accordingly, MAs expressed the desire for more recognition from politicians. In Germany, many health care professionals received a governmental "corona bonus"- However, this was not the case for MAs. They hoped that politicians will take their commitment into account and appreciate it more in the future:

> MA-03: "Yes, somehow more recognition from politicians perhaps, that there would be some kind of […] co-payments or - that's somehow always the case with politics, that they say, well, it's not just geriatric nurses, medical assistants have also had a hard time, I would like to see that somehow, in the future, that they are also kept more in mind."

It seems that there is a need to recognize other groups within the medical workforce and not just focus on specific professions. Without the appreciation required above, MAs develop strategies to deal with such situations. This can involve resilience but also turning away from the current profession, as shown in the next section.

## Resilience vs. migration

During the Covid-19 pandemic, many MAs have taken the opportunity to reflect on their professions. The challenges and changes coming along with this unforeseen situation led them to reflect on the meaning of their profession and how they perceive their work:

> MA-06: " And during that time, I came to the conclusion that I wanted to do something else. I then started further training and - yes, I've now become a freelancer and will soon be working a little less as a medical assistant, I can honestly tell you."

Another MA described how patients were often abusive. She emphasized the stress this placed on medical staff and expressed concerns about recommending the medical assisting profession to others due to this stress:

> MA-14: "The patients are simply really abusive, both in their choice of words and in the tone of their voice. And these are all things that can't (-) go on like this. We've been listening to this for a year now, and it's just miserable. I spoke to a few young girls at our sports club, and I said 'You can be anything, just don't become a MA. For God's sake, find something else'. [...] You can't recommend this job to anyone with a clear conscience. It's no longer possible."

Another MA was concerned about the risk for her own health:

> MA-14: "Hm, I thought it was - yes, very, very threatening. And then one of our supervisors said, 'Well, if you've taken on the job as a medical assistant - I learned that 30 years ago - you had to know that you'd get into a situation like this'. No, I couldn't have known that 30 years ago And I also told him 'If I would have wanted that, I would have gone to the Congo and cared for Ebola patients there'. But I don't want that kind of thing."

While supervision might be a means to foster resilience, this MA felt not taken seriously in her sorrows. Contrary, a strong team and informal communication between the colleges were helpful in coping with the situation according to one MA. Despite the extreme situations, some teams were able to come even closer together. Solidarity was mentioned as an important element in difficult times:

> MA-07: "And that a good team is important, that you can get along with each other even in difficult situations. Because if you have someone who says 'Oh come on, we'll get through it' when things aren't going so well, that makes a big difference. It's not like everyone is a lone fighter and nobody in the team cares how the other person is doing."

> MA-01: "We stick together even more than before. Quite simply, we have become even more united as a result of the whole thing because we absolutely know how we can rely on each other. Even in these extreme situations that have arisen."

Overall, the quotes show the resilience of MAs in times of stress and uncertainty. At the same time, individual considerations for career changes due to the pressures in the medical field and resentfulness were presented.

### Needs and wishes in everyday work

It was highlighted that in uncertain situations, such as the emergence of new vaccines during the pandemic, a point of support would have been important:

> MA-06: "[...] having such a support point would have been nice. When things like that happen, there really is someone somewhere who can answer questions explicitly. Because even our doctors couldn't say 'This vaccine is for you or that one', it was all too new and too recent and no one really knew that, right?"

The MA urged more targeted support and suggested working participatory with practices to gain direct insight. Politicians were encouraged to experience day-to-day practice life to understand the feasible challenges and provide realistic support:

> MA-11: "You don't even see what services the MAs provide at the front, always at the front, they are always - yes, the cannon fodder, to put it bluntly (laughing), right? Every sick person, every patient, whether corona yes/no, arrives here first, right? And to handle all that, to manage it - you had to be very flexible; and I would have wished for a little more recognition from the side - yes, maybe a little more recognition."

> MA-04: "Perhaps they should simply take a look at everyday life (-) in a completely normal, ordinary practice so that they can have a say and so that they know what they are talking about: What does everyday life look like? How is it structured? Can I do that? Can I still get funding, or can I support them in some other way?"

The quotes show the need of MAs for clear communication, targeted support, and recognition for their daily efforts. It is pointed out that politicians and decision-makers should understand more clearly what everyday life in medical facilities looks like to provide appropriate support.

### Discussion

The presented qualitative study evaluated the MAs perceptions during the peak of the Covid-19 pandemic in Germany and will be discussed below with already published articles. It gives

us deep insights into the MAs-patient relationship during the pandemic and addresses several ethical issues.

In general, it became apparent that the daily work of MAs has been energy-sapping during the pandemic. The MAs reported on the chaos in the daily organization, the anxieties of all involved and their collective trauma. It was not only the statements not kept by politicians and the unstructured, often equivocal communication by the media that caused demotivation in the everyday work of the MAs, but also the increased expectations and associated frustration when patients' expectations were not met during the Covid-19 pandemic [15]. Patients demanded more and were more impatient if their expectations could not be met quickly enough [1,15].

The high expectations of patients were also reflected in their egocentric behavior and disapproval of patients when MAs did not immediately provide all information and could not answer all questions, even if they could also only obtain their information from the media [1]. Furthermore, the MAs also had to deal with demanding attitude from patients and stolen material, so some practices attempted to employ security staff, as reported above. The behavior of the patients put pressure on the MAs, but it should be considered that the patients were also exposed to enormous uncertainties. It must be kept in mind that the patients also showed fear of the emerging virus and often did not demand to be prioritized out of malice [16,17]. The pandemic was a threatening time, and people wanted to protect themselves and their relatives. In addition, patients had to deal with the possible consequences of a Covid-19 infection, which for many were not yet predictable.

The working conditions of MAs have already been problematic due to their high workload and different tasks with overtime work and low salaries [1,18], but with the onset of the Covid-19 pandemic and especially the vaccination campaign, they worked a lot of overtime that was not compensated [1,15]. Although they represented a significant part of the protective wall that shielded the clinics from being overloaded, they criticized that they were simply not appreciated enough, unlike the nursing staff, who were praised in the media [19–21]. Studies also show that other healthcare workers, such as nurses, doctors and especially primary contact personnel who have first contact with patients, are also affected by stress, an increased workload and difficult patient contacts [22–24]. Nevertheless, the MAs criticized in the interviews that they felt "not seen".

The increased workload, which is intensified by a lack of appreciation, is particularly detrimental. Job satisfaction is influenced by working hours and high workload as well as lack of appreciation and insufficient salary [25–28]. In such stressful situations, healthcare staff tend to be absent due to illness more often or are completely overwhelmed with their work, as previous studies on epidemics have shown [29]. At the beginning of the pandemic, 10.9% of healthcare workers (especially doctors and nurses) were already considering changing careers [27,29,30]. It is uncertain what the situation was explicitly like for MAs. However, according to our interviews, MAs are considering career changes due to the challenges of their work during the pandemic.

To avoid this situation, employers need to recognize stressed employees and offer interventions that at best have been documented previously in a kind of crisis plan. Many practices were able to learn from the past situation and draw up individual recommendations for action. In the future, extraordinary stressful situations may continue to occur, for which employers and MAs should prepare in collaboration. Appreciation from employers and support from the management are also important factors in counteracting work overload [29]. Our study also showed that appreciative interaction between employees and an overall supportive working environment has an encouraging influence and can make a big difference in everyday work. This can create a sense of togetherness and prevent all

employees from struggling alone, thus creating resilience and enabling a rapid recovery of mental health [29].

In crises, centralized information points must be created where patients can obtain information so that their fears and concerns can be allayed. In addition, politicians and the media should only release information as soon as it is considered reliable. Releasing information prematurely, which may have to be revised later, unsettles the population, and can lead to anxiety and frustration, which is later taken out on the MAs. Most recently, the many strikes by MAs under the leadership of the Association of Medical Professions (Verband Medizinischer Fachberufe e.V.) in Germany, showed how important reasonable framework conditions are to MAs within their profession [31]. Many MAs pointing out demanding and aggressive patients in social media also shows how society's attitude should change so that general practices do not face a flood of layoffs in the future.

Recommendations for future improvements include enhancing support for MAs, recognizing their contributions to the media, and fostering collaborative efforts between practitioners and policymakers to gain insights into the daily challenges of medical facilities. Additionally, creating a more supportive and appreciative work environment, with a focus on teamwork and resilience, can contribute to the well-being of MAs.

## Conclusion

The results of our study shed light on the significant challenges faced by MAs during the Covid-19 pandemic, particularly in general practices.

The pandemic exacerbated existing difficulties in the daily work of MAs, with increased demands, unstructured communication from the media, and a surge in disrespectful behavior from patients. Patients exhibited more impatience and egocentric behavior, adding to the burdens faced by MAs. Additionally, incidents of abusive behavior and theft of materials further strained the work environment, prompting some practices to consider security measures. Despite their critical role in maintaining healthcare services, MAs felt undervalued and unappreciated, especially in comparison to other healthcare professions. The lack of appreciation, coupled with increased workloads and stress, contributed to concerns about burnout and career changes or participating in retraining measures among MAs.

Our PIPER study highlighted the broader societal shift towards self-centered behavior, with patients displaying increased demands and impatience, often fueled by misinformation from the media. The study underscored the importance of acknowledging and addressing the psychological burden faced by MAs, as their well-being directly influences the quality of patient care.

To address these challenges, employers and policymakers must recognize the importance of supporting MAs and provide intervention strategies to mitigate stress and workload. Centralized information points for patients and improved communication from politicians and the media can help alleviate uncertainties and reduce tensions in medical settings.

Efforts to promote a supportive and appreciative working environment and initiatives to encourage collaboration between General Practitioners and policy makers are essential to improving the quality of working conditions for MAs. By acknowledging their contributions, providing targeted support, and promoting resilience, we can ensure the continued effectiveness of healthcare systems and the overall well-being of MAs.

The following three main points could represent concrete measures to address the challenges mentioned in this article:

1) Legal requirements for mental health in the workplace: Regular and mandatory crisis supervision could help MAs deal with difficult situations and strengthen teamwork.

2) Creating a legal framework for better patient communication: It could be required by law that all health information is communicated clearly, reliably, and transparently during a crisis to avoid uncertainty. Control bodies should be implemented. Policies to educate media representatives and policymakers on the impact of misinformation on public health could help reduce excessive demands and aggressive patient behavior.

3) Mandatory safety precautions in medical facilities: Legislation could be introduced to require that medical facilities implement appropriate security measures (e.g., security guards or deliberate locking away of materials) to protect staff from physical violence and theft, as has been reported during the pandemic and has already been reported by MAs interviewed.

The role of practice owners and associations of statutory health insurance physicians should not be underestimated when it comes to shaping the working conditions of medical assistants. Practice owners are directly responsible for determining how the working environment and conditions look on a day-to-day basis. Through fair pay, regular training, and psychological support, they can help to ensure that employees feel valued and do not burn out. At a political level, the associations of statutory health insurance physicians have the task of advocating for the rights and interests of MAs, particularly regarding working hours, overtime pay, and the quality of training opportunities. Close cooperation between practice owners and statutory health insurance physicians' associations can achieve a sustainable improvement in working conditions that benefits both the MAs and the patients.

## Strengths and limitations

A significant strength of this study is the timing of the interviews. Between April and September 2021, the first wave of Covid-19-vaccination was conducted in general practices. This allowed us to learn just in time about the daily work and struggles of MAs as well as their experiences with aggressive patients, which gave us deep insights into MA-Patient relationship during the pandemic. The content of the interviews conducted during this period was extensive and detailed, as the MAs generously shared their experiences and unfiltered emotions despite their increased workload in the practices.

However, a limitation of this study relates to the fact that only MAs who volunteered to participate in the study were interviewed. Therefore, it can be assumed that only individuals who had an interest in the thematic content and were willing to talk about it shared their insights.

## Supporting information

**S1 File. Translated interview guideline.**
(DOCX)

**S2 File. Translated coding tree.**
(DOCX)

**S3 File. COREQ Checklist.**
(PDF)

## Author contributions

**Conceptualization:** Anastasia Suslow, Kathrin Schlößler, Nino Chikhradze, Romy Lauer, Horst Christian Vollmar, Ina Carola Otte.

**Data curation:** Anastasia Suslow.

**Formal analysis:** Anastasia Suslow, Kathrin Schlößler, Nino Chikhradze.

**Funding acquisition:** Horst Christian Vollmar, Ina Carola Otte.

**Methodology:** Anastasia Suslow, Kathrin Schlößler.

**Project administration:** Anastasia Suslow, Horst Christian Vollmar, Ina Carola Otte.

**Supervision:** Horst Christian Vollmar, Ina Carola Otte.

**Writing – original draft:** Anastasia Suslow.

**Writing – review & editing:** Kathrin Schlößler, Nino Chikhradze, Romy Lauer, Michael Pentzek, Achim Mortsiefer, Horst Christian Vollmar, Ina Carola Otte.

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
