## [Decision Letter · Decision Letter 0]

14 Jan 2025

PONE-D-24-24572Should I stay or should I go – Medical assistants´ experiences and coping with patient demand and lack of appreciation during the Covid-19 pandemicPLOS ONE

Dear Dr. Suslow, Thank you for submitting your manuscript to PLOS ONE. After careful consideration, we feel that it has merit but does not fully meet PLOS ONE’s publication criteria as it currently stands. Therefore, we invite you to submit a revised version of the manuscript that addresses the points raised during the review process. Please submit your revised manuscript by Feb 28 2025 11:59PM. If you will need more time than this to complete your revisions, please reply to this message or contact the journal office at plosone@plos.org . Please include the following items when submitting your revised manuscript:

We look forward to receiving your revised manuscript.

Kind regards,

Othman A. Alfuqaha, Ph.D.

Academic Editor

PLOS ONE

Journal Requirements:

“I have read the journal's policy and the authors of this manuscript have the following competing interests: Kathrin Schlößler is an employed doctor in a general practitioner's practice from whose practice MA were interviewed for this study. As an employed GP, she also worked as a trial physician in a study under the German Drug Law (sponsored by Lilly). She received her regular salary as an employed doctor and received no compensation or expense allowance from the sponsor.”

3. In the online submission form, you indicated that [Insert text from online submission form here].

Additional Editor Comments:

Dear author's,

Thank you for submitting your manuscript to PLOS ONE. After careful consideration of the reviewers’ feedback.

Both reviewers acknowledged the significance and relevance of your study and commended the manuscript for its well-structured and well-written presentation. They also provided constructive suggestions to further enhance the clarity and impact of your work.

Please follow the reviewers' comments that will need to be addressed in your revised submission:

Abstract

Revise the final sentence in the results section, as it reads more like a conclusion.

Methods

Ensure consistency with the COREQ checklist.

Provide greater clarity on the data collection process, including how topics were addressed in interviews and how the coding process was conducted.

Merge overlapping headings (e.g., "Instruments" with "Data Collection"; "Evaluation" with "Analysis Process").

Include details about interview length and who conducted the interviews.

Relocate certain content currently in the results section (e.g., lines 145–150) to the methods section.

Results

Shorten and focus on the essential findings, avoiding redundancy.

Clarify the distinction between participant statements and the authors’ interpretations.

Provide an overview of categories, including a code tree.

Avoid excessive quotations and unnecessary repetition.

Discussion and Conclusion

Ensure any claims about comparisons with pre-pandemic work are supported by results.

Expand the discussion to include relevant literature, especially on healthcare professions with similar challenges.

Highlight actionable measures, including legislative-level improvements, that could address the identified issues.

Additional Points

Provide clarification on the roles and job descriptions of MFAs.

Include any relevant international studies on the workload of medical assistants during the study period.

Discuss the roles of practice owners and professional associations (e.g., KVen) in shaping working conditions.

Indicate whether participants were compensated for their interviews.

These revisions are expected to enhance the clarity and depth of your manuscript while maintaining its focus and relevance.

We look forward to receiving your revised manuscript.

Regards,

Othman A. Alfuqaha

Academic Editor

PLOS ONE

Reviewers' comments:

Reviewer's Responses to Questions

**Comments to the Author**

1. Is the manuscript technically sound, and do the data support the conclusions?

Reviewer #1: Yes

Reviewer #2: Yes

2. Has the statistical analysis been performed appropriately and rigorously? 

Reviewer #1: Yes

Reviewer #2: N/A

3. Have the authors made all data underlying the findings in their manuscript fully available?

Reviewer #1: Yes

Reviewer #2: Yes

4. Is the manuscript presented in an intelligible fashion and written in standard English?

Reviewer #1: Yes

Reviewer #2: Yes

5. Review Comments to the Author

Reviewer #1: The manuscript presented can be valued very positively and follows a clear and valid rationale. The methodological approach is comprehensible and suitable for answering the research questions. I recommend the following very minor revisions for additional enhancement:

Background:

(p.3, 64-66): The job description and tasks of MFAs are highly dependent on whether they are an MFA without an additional designation, a VERAH/EVA/NÄPA, or a Physician Assistant. In the past, study programs for physician assistants were also called “Medizinische Assistenz” in Germany. It is also not clear that MA is a training occupation ("Ausbildungsberuf"). This part could be more precise, as it concerns the target group of the work. This also applies to the description of the participants in the results section.

Are there any international studies on the workload of medical assistants during the study period in GP practices? If they exist, they should be added

Methods:

The methodological approach is transparent and target-oriented. The number of 21 interviews is also sufficient. If the interview participants were counted for compensation, I would recommend indicating this.

Results:

Very interesting and relevant results. I feel that the role of the practice owners and possibly also the associations of statutory health insurance physicians (KVen) is a little bit neglected. Perhaps they were not mentioned by the MAs? Not only “politicians” are decision-makers, but also the joint self-administration (Gemeinsame Selbstverwaltung) and the KVen. Even when it comes to financial compensation for overtime hours or a “corona bonus”, this is not the task of politicians, but of the practice owners. They have also been able to benefit financially, at least in part, and could have passed on some of the profits to the practice staff.

Discussion & Conclusion

Very purposeful discussion. The role of employers is also explained here. Working conditions of MAs can be shaped to a large extent by professional policy (Berufspolitik) and practice owners. Additional measures that may be necessary at the legislative level should be mentioned if they exist.

Reviewer #2: Thank you very much for the manuscript. It focuses on an interesting topic and shows how important the information of the analysed group is. I read it with great interest. It is well structured and well written. There are many interesting insights into the perceptions of the MA.

Nevertheless, the manuscript has some shortcomings that I would like to suggest to correct or change.

- In the abstract, the last sentence of the results seems more like a conclusion to me.

- The background is well summarised and well written.

- Methods: The information in the COREQ Checklist does not seem to match the manuscript.

o The data collection procedure remains somewhat unclear to me. How were the topics addressed in the interview? Terms such as narrative, semi-structured (semi-structured interview guideline) are used. What exactly was collected and how, i.e. with which questions?

o Table 1 shows the topics in the interviews. Were the topics known in advance? How exactly were the topics addressed? Were they all addressed in each interview?

o line 100: the heading here is ‘Instruments’ - data collection would be more appropriate. (I would like to recommend that the two sections be dealt with under one heading (lines 100 - 116).)

o line 117: The heading here is ‘Evaluation’ – better ‘Analysis process’

o In my view, information is missing here that then appears in the results (e.g. the length of the interviews, who conducted the interviews?) Please check again with the COREQ-Checklist.

o line 145-150: In my opinion, the content belongs in the methods section - it says that it is a semi-structured interview guideline and that it was coded according to this, so what is written here seems implausible to me.

o I would like to be introduced to the code tree.

o In principle, lines 145 - 165 seem to me to belong more in the method section. Method and results are too mixed up here. Please separate them more clearly.

- Results: At the beginning of the results section (line 145-165 too), please give a very brief introduction (also in the form of a table/overview) as to how the results will be presented, but do not give a substantive introduction to the topics (i.e. this does not have to be justified).

o The results are very lengthy, maybe not everything has to be presented in every detail, please always concentrate on essentials. It is normal, that the categories can frequently not be separated accurately. Please avoid repeating yourself (not every statement needs a citation as proof!).! In view of this, the results should be reduced.

o In some places, the quotations are simply strung together and shed light on different aspects; the reader should be taken by the hand.

o And with the quotations and their interpretations, the question sometimes arises: is the method of analysis really a content analysis? The distinction between content-analytical reproduction of the content and additional interpretations is difficult to distinguish...the latter, however, belongs above all in the discussion

o e.g. for unclear interpretation level: line 168-169 “During the pandemic, MAs made experiences with demanding patients and social egoism that frightened them and let them question their view of the world and humanity as a whole.“ Sounds like that's the truth. It is unclear whether this is the case, whether the MFAs saw it this way and reported it this way, or whether it is the authors' interpretation. For a qualitative presentation, ‘it seems so’ can also be used.

o Overall, it is always noticeable that as a reader I would like more information about the categories and main categories as a whole.

o Please tighten up the results section so that the text and then the example do not contain the same information. Do not duplicate quotations or avoid similar ones, it is sufficient if this is illustrated.

- conclusion: Line 442 “The pandemic exacerbated existing difficulties in the daily work of Mas”: Can a comparison be drawn with the work before the pandemic based on the interviews? I didn't see this so much in the results. Please adjust, if necessary.

- In addition to the extensive results, the discussion seems very straightforward. There is probably little literature on this directly, but there certainly is on the topics analysed in relation to other healthcare professions.

- As far as the limitations are concerned, the following would occur to me: the quotations are already very impressive, perhaps a different type of analysis would have been conceivable in order to do justice to the data material?

6. PLOS authors have the option to publish the peer review history of their article (what does this mean? ). If published, this will include your full peer review and any attached files.

**Do you want your identity to be public for this peer review?** For information about this choice, including consent withdrawal, please see our Privacy Policy .

Reviewer #1: No

Reviewer #2: No

---

## [Author Response · Author response to Decision Letter 0]

21 Feb 2025

Dear Prof. Chenette,

Thank you very much for your recommendation and the helpful reviewer comments.

Please note that we have made every effort to make the requested changes.

Below we have set out our approach in a point-to-point reply.

We thank you for your time and would appreciate the opportunity for publication.

Yours sincerely,

Anastasia Suslow for the authors

We have checked the style and have endeavored to meet it.

“I have read the journal's policy and the authors of this manuscript have the following competing interests: Kathrin Schlößler is an employed doctor in a general practitioner's practice from whose practice MA were interviewed for this study. As an employed GP, she also worked as a trial physician in a study under the German Drug Law (sponsored by Lilly). She received her regular salary as an employed doctor and received no compensation or expense allowance from the sponsor.”

Please include your updated Competing Interests statement in your cover letter; we will change the online submission form on your behalf. We have added the suggested statement.

3. 3. In the online submission form, you indicated that [Insert text from online submission form here].

This policy applies to all data except where public deposition would breach compliance with the protocol approved by your research ethics board. If your data cannot be made publicly available for ethical or legal reasons (e.g., public availability would compromise patient privacy), please explain your reasons on resubmission and your exemption request will be escalated for approval. We have translated the guidelines and code tree and are adding them as supplementary material. We hope you understand that we are unable to make the interview transcripts available due to data protection regulations.

4. 4. Please amend your list of authors on the manuscript to ensure that each author is linked to an affiliation. Authors’ affiliations should reflect the institution where the work was done (if authors moved subsequently, you can also list the new affiliation stating “current affiliation:….” as necessary). We have extended RL's, NC’s and ICO’s affiliation as they have changed departments in the meantime.

5. 5. Your ethics statement should only appear in the Methods section of your manuscript. If your ethics statement is written in any section besides the Methods, please delete it from any other section. We appreciate the advice! We actually forgot to delete one of the statements. Now it only appears in the method section.

6. 6. Please review your reference list to ensure that it is complete and correct. If you have cited papers that have been retracted, please include the rationale for doing so in the manuscript text, or remove these references and replace them with relevant current references. Any changes to the reference list should be mentioned in the rebuttal letter that accompanies your revised manuscript. If you need to cite a retracted article, indicate the article’s retracted status in the References list and also include a citation and full reference for the retraction notice Many thanks for this advice. We have checked the references and updated [5] and [12] with regard to the new editions. In addition, [18], [22], [23] and [24] have been added at the request of the reviewers. Unfortunately, we were unable to identify articles that had been retracted. Could you please tell us specifically which articles these are?

7. Thank you for submitting your manuscript to PLOS ONE. After careful consideration of the reviewers’ feedback.

Both reviewers acknowledged the significance and relevance of your study and commended the manuscript for its well-structured and well-written presentation. They also provided constructive suggestions to further enhance the clarity and impact of your work.

Please follow the reviewers' comments that will need to be addressed in your revised submission:

Abstract

Revise the final sentence in the results section, as it reads more like a conclusion. We have changed the last sentence to reflect this statement in the interviews.

8. Methods

Ensure consistency with the COREQ checklist. We have also adapted the COREQ checklist in line with the changes made and hope that it is now consistent.

9. Provide greater clarity on the data collection process, including how topics were addressed in interviews and how the coding process was conducted. We have expanded our methods section and are hopeful that our approach has been clarified.

10. Merge overlapping headings (e.g., "Instruments" with "Data Collection"; "Evaluation" with "Analysis Process"). We merged the headings.

11. Include details about interview length and who conducted the interviews. We have added the information about who conducted the interviews and how long they lasted.

12. Relocate certain content currently in the results section (e.g., lines 145–150) to the methods section. We are grateful for the comment and relocated the content mentioned here to the methods section.

13. Results

Shorten and focus on the essential findings, avoiding redundancy. We have shortened the results section considerably and thus hope that we have been able to eliminate repetitions and get straight to the point.

14. Clarify the distinction between participant statements and the authors’ interpretations. We have added a statement to clarify which quotes are from the participants and which sections are interpretations by the authors.

15. Provide an overview of categories, including a code tree. We have translated the code tree and added it as an attachment.

16. Avoid excessive quotations and unnecessary repetition. Please see 13)

17. Discussion and Conclusion

Ensure any claims about comparisons with pre-pandemic work are supported by results. To illustrate the comparison of the workload before and during the pandemic, we have included a quote in the results in which an MA says that she had to increase her working hours and added evidence of the previous workload in the discussion.

18. Expand the discussion to include relevant literature, especially on healthcare professions with similar challenges. We have added studies with other healthcare professions to the discussion and hope to show why our research is relevant.

19. Highlight actionable measures, including legislative-level improvements, that could address the identified issues. We have added the following measures to address the identified issues in our article:

1) Legal requirements for mental health in the workplace:

Regular and mandatory crisis supervision could help MAs deal with difficult situations and strengthen teamwork.

2) Creating a legal framework for better patient communication:

It could be required by law that all health information is communicated clearly, reliably, and transparently during a crisis to avoid uncertainty. Policies to educate media representatives and policymakers on the impact of misinformation on public health could help reduce excessive demands and aggressive patient behavior.

3) Mandatory safety precautions in medical facilities:

Legislation could be introduced to require that medical facilities implement appropriate security measures (e.g., security guards or deliberate locking away of materials) to protect staff from physical violence and theft, as has been reported during the pandemic and has already been reported by MAs interviewed.

20. Additional Points

Provide clarification on the roles and job descriptions of MFAs. Please see 24)

21. Include any relevant international studies on the workload of medical assistants during the study period. Unfortunately, we could not find any relevant international literature on the working conditions of MAs during the pandemic. Nevertheless, we have added literature on healthcare workers to provide some comparison.

22. Discuss the roles of practice owners and professional associations (e.g., KVen) in shaping working conditions. In order to supplement our article with far-reaching possibilities and proposed solutions, we have added a section on the role of GPs and KVen:

The role of practice owners and associations of statutory health insurance physicians should not be underestimated when it comes to shaping the working conditions of medical assistants. Practice owners are directly responsible for determining how the working environment and conditions look on a day-to-day basis. Through fair pay, regular training, and psychological support, they can help to ensure that employees feel valued and do not burn out. At a political level, the associations of statutory health insurance physicians have the task of advocating for the rights and interests of MAs, particularly with regard to working hours, overtime pay, and the quality of training opportunities. Close cooperation between practice owners and statutory health insurance physicians' associations can achieve a sustainable improvement in working conditions that benefits both the MAs and the patients.

23. Indicate whether participants were compensated for their interviews.

These revisions are expected to enhance the clarity and depth of your manuscript while maintaining its focus and relevance.

We look forward to receiving your revised manuscript. We have added the information that the MAs did not receive any compensation for the interviews.

24. Reviewer #1: The manuscript presented can be valued very positively and follows a clear and valid rationale. The methodological approach is comprehensible and suitable for answering the research questions. I recommend the following very minor revisions for additional enhancement:

Background:

(p.3, 64-66): The job description and tasks of MFAs are highly dependent on whether they are an MFA without an additional designation, a VERAH/EVA/NÄPA, or a Physician Assistant. In the past, study programs for physician assistants were also called “Medizinische Assistenz” in Germany. It is also not clear that MA is a training occupation ("Ausbildungsberuf"). This part could be more precise, as it concerns the target group of the work. This also applies to the description of the participants in the results section. We have added a few additions and hope to be able to provide international readers with a better understanding of how MAs operate in Germany.

25. Methods:

The methodological approach is transparent and target-oriented. The number of 21 interviews is also sufficient. If the interview participants were counted for compensation, I would recommend indicating this. Thank you very much for this feedback! We have added a statement stating that the MAs did not receive any compensation.

26. Results:

Very interesting and relevant results. I feel that the role of the practice owners and possibly also the associations of statutory health insurance physicians (KVen) is a little bit neglected. Perhaps they were not mentioned by the MAs? Not only “politicians” are decision-makers, but also the joint self-administration (Gemeinsame Selbstverwaltung) and the KVen. Even when it comes to financial compensation for overtime hours or a “corona bonus”, this is not the task of politicians, but of the practice owners. They have also been able to benefit financially, at least in part, and could have passed on some of the profits to the practice staff. That is an interesting aspect. Unfortunately, these factors were not addressed in our interviews. Overall, the politicians and the media were actually criticized. This can probably be attributed to the fact that the interviews were conducted during the vaccination organization and politicians and the media were the first sources of anger. Just as the patients vented their frustration to the MAs.

27. Discussion & Conclusion

Very purposeful discussion. The role of employers is also explained here. Working conditions of MAs can be shaped to a large extent by professional policy (Berufspolitik) and practice owners. Additional measures that may be necessary at the legislative level should be mentioned if they exist. Please see 19) and 22)

28. Reviewer #2: Thank you very much for the manuscript. It focuses on an interesting topic and shows how important the information of the analysed group is. I read it with great interest. It is well structured and well written. There are many interesting insights into the perceptions of the MA.

Nevertheless, the manuscript has some shortcomings that I would like to suggest to correct or change.

29. - In the abstract, the last sentence of the results seems more like a conclusion to me. We have made an effort to change the last sentence so that it fits better in the results section. See also 7)

30. The background is well summarised and well written. Thank you very much.

31. Methods: The information in the COREQ Checklist does not seem to match the manuscript. We have also adapted the COREQ checklist in line with the changes made and hope that it is now consistent.

32. The data collection procedure remains somewhat unclear to me. How were the topics addressed in the interview? Terms such as narrative, semi-structured (semi-structured interview guideline) are used. What exactly was collected and how, i.e. with which questions? We have added the interview guide as an appendix and hope that this will make our approach clearer. We also added some more detailed information in the method section. See also 9)

33. Table 1 shows the topics in the interviews. Were the topics known in advance? How exactly were the topics addressed? Were they all addressed in each interview? We have decided to translate the interview guide and attach it as an appendix to provide a better understanding of our approach. We additionally added some more information about the way the interviews were conducted (see also 9)). We hope to be able to clarify the question in this way. The participants had no prior insight into the questions.

34. line 100: the heading here is ‘Instruments’ - data collection would be more appropriate. (I would like to recommend that the two sections be dealt with under one heading (lines 100 - 116).) Thank you for this advice. We have combined the two sections.

35. line 117: The heading here is ‘Evaluation’ – better ‘Analysis process’ We have also followed this advice and hope that the subheadings are now more appropriate.

36. In my view, information is missing here that then appears in the results (e.g. the length of the interviews, who conducted the interviews?) Please check again with the COREQ-Checklist. We have added the information about who conducted the interviews and how long they lasted.

37. line 145-150: In my opinion, the content belongs in the methods section - it says that it is a semi-structured interview guideline and that it was coded according to this, so what is written here seems implausible to me. Please see 12).

38. I would like to be introduced to the code tree. We have translated the code tree and added it as an attachment.

39. In principle, lines 145 - 165 seem to me to belong more in the method section. Method and results are too mixed

---

## [Editor Report · Decision Letter 1]

27 Feb 2025

Should I stay or should I go – Medical assistants´ experiences and coping with patient demand and lack of appreciation during the Covid-19 pandemic

PONE-D-24-24572R1

Dear Dr.

<table border="0" cellpadding="0" cellspacing="0" class="datatable3" style="border-collapse: collapse; width: 678px; line-height: 14px; color: rgb(0, 0, 51); font-family: verdana, geneva, arial, helvetica, sans-serif; font-size: 11.2px;"> <tbody> <tr style="background-color: rgb(244, 244, 244);"> <td style="padding: 3px; border: 1px solid rgb(255, 255, 255);">Anastasia Suslow,</td> </tr> <tr style="background-color: rgb(244, 244, 244);"> <td style="padding: 3px; border: 1px solid rgb(255, 255, 255); width: 196.094px;"> </td> </tr> </tbody></table>

We’re pleased to inform you that your manuscript has been judged scientifically suitable for publication and will be formally accepted for publication once it meets all outstanding technical requirements.

Kind regards,

Othman A. Alfuqaha, Ph.D.

Academic Editor

PLOS ONE

Additional Editor Comments (optional):

Dear Authors,

I am pleased with your response to the reviewers' comments and appreciate the effort you have put into addressing them thoroughly. I would like to take this opportunity to commend you on your valuable contributions to this research. Congratulations on your achievement, and I wish you continued success in your academic endeavors.

Best regards,
---

## [Editor Report · Acceptance letter]

PONE-D-24-24572R1

PLOS ONE

Dear Dr. Suslow,

I'm pleased to inform you that your manuscript has been deemed suitable for publication in PLOS ONE. Congratulations! Your manuscript is now being handed over to our production team.

Kind regards,

on behalf of

Dr. Othman A. Alfuqaha

Academic Editor

PLOS ONE